# Comment on Lombardo et al. The Impact of Insulin-Induced Lipodystrophy on Glycemic Variability in Pediatric Patients with Type 1 Diabetes. *Children* 2022, *9*, 1087

**DOI:** 10.3390/children12020111

**Published:** 2025-01-21

**Authors:** Timothy P. Foster, Desmond Schatz

**Affiliations:** Department of Pediatrics, Division of Endocrinology, College of Medicine, University of Florida, Gainesville, FL 32608, USA

As Type 1 diabetes pediatric physician-scientists, we read the publication “The Impact of Insulin-Induced Lipodystrophy on Glycemic Variability in Pediatric Patients with Type 1 Diabetes” by Lombardo et al. with interest [1]. Insulin-induced lipohypertrophy (LH) is a localized lipodystrophy that results from repeated subcutaneous injections at the same site. While diabetes technology has rapidly improved over the past decade, lipodystrophy still represents a significant barrier to stable glycemic variability and increased risk of severe hypoglycemia by altering the absorption of insulin. This study adds to decades of supportive research that LH identification and injection site rotation are vital aspects of diabetes education and clinical evaluations.

The authors performed a cross-sectional study evaluating lipodystrophy in 212 pediatric patients with type 1 diabetes. LH was common among their participants, with 94/212 (44.3%) having at least one lesion. Two patients were noted to have lipoatrophy. In contrast to other studies, Lombardo et al. found no statistically significant differences in BMI, HbA1c, daily insulin use, injection vs. infusion, or diabetes duration between those with and without lipodystrophy [2,3]. However, they did note a higher coefficient of variation (CV) in continuous or flash glucose monitoring (CGM, FGM) in those with sites of lipodystrophy (*p* = 0.036). The authors rightfully emphasize that this represents a negative impact on glycemic variability, which independently increases risks of diabetes complications [1].

This study also importantly highlights the differences in insulin site rotation and lipodystrophy awareness on lipodystrophy. There was no reported injection site variation in 19.8% of those with lipodystrophy compared to 10.3% of those without (*p* = 0.05), and there was over 3.6 times the rate of lipodystrophy unawareness in those that had lipodystrophy (*p* = 0.005). Recent studies have described dermatologic changes at insulin infusion sites, including fibrosis and inflammation, with increased IGF-1 and TGF-β at both current and recovering (3–5-day prior) pump sites [4]. These histopathologic changes provide additional evidence that subcutaneous sites of insulin administration require significant recovery time before reuse. As such, education should be expanded beyond generalized recommendations for injection site rotation, and should emphasize avoiding the repeated use of a previous injection/pump site for as long as possible.

Unfortunately, the frequency of injection site assessments and re-education is low in many children and adolescents with T1D. Increased awareness of the importance of LH has led to the development of international guidelines for managing injection areas and for detecting LH [5]. Our clinic visits must incorporate patient-centered re-education on insulin site variation and communication of awareness/risks of LH to our patients. In-person physical examination is also necessary, as lipodystrophy assessment and technique retraining can improve HbA1c and daily insulin use [6]. This is especially paramount to remember in a time when telemedicine can easily replace face-to-face visits. While virtual visits can improve access to care, traditional in-office visits are still valuable in providing optimal diabetes care.

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
