# Peer review of "Comment on Lombardo et al. The Impact of Insulin-Induced Lipodystrophy on Glycemic Variability in Pediatric Patients with Type 1 Diabetes. Children 2022, 9, 1087"

_children, 2025, doi:10.3390/children12020111_

Round 1
Reviewer 1 Report (New Reviewer)
Comments and Suggestions for Authors
This study reports a higher coefficient of variation (CV) observed in continuous or flash glucose monitoring (CGM, FGM) among individuals with lipodystrophy at injection sites, demonstrating its adverse effects on glycemic variability. It underscores the critical importance of insulin site rotation and raising awareness about lipodystrophy.
The text is clear, well-written, and effectively conveys its message.
This manuscript is a resubmission of an earlier submission. The following is a list of the peer review reports and author responses from that submission.
Round 1
Reviewer 1 Report
Comments and Suggestions for Authors
Most of the part of this comment repeats the results of the paper with title "The Impact of Insulin-Induced Lipodystrophy on Glycemic Variability in Pediatric Patients with Type 1 Diabetes" in order to conclude that repeated injections in the same injection site should be avoided for as long as possible and that face to face clinics should be performed in order to monitor injection sites.
It is good to remind practitioners to keep having face to face clinics and to keep advising their patients to change injection sites. However, this comment doesn't add much our knowledge so far.